# Mental Adjustment as a Predictor of Comprehensive Quality of Life Outcome among Patients with Terminal Cancer

**DOI:** 10.3390/ijerph18094926

**Published:** 2021-05-05

**Authors:** Li-Fang Chang, Chi-Kang Lin, Li-Fen Wu, Ching-Liang Ho, Yi-Ling Lu, Hsueh-Hsing Pan

**Affiliations:** 1Department of Nursing, Tri-Service General Hospital, Taipei 11490, Taiwan; fang_niki@mail.ndmctsgh.edu.tw (L.-F.C.); wulifen@mail.ndmctsgh.edu.tw (L.-F.W.); 607209027@mail.ndmctsgh.edu.tw (Y.-L.L.); 2Graduate Institute of Medical Sciences, National Defense Medical Center, Taipei 11490, Taiwan; 3Department of Gynecology and Obstetrics, Tri-Service General Hospital, National Defense Medical Center, Taipei 11490, Taiwan; kung568@mail.ndmctsgh.edu.tw; 4Division of Hematology and Oncology, Tri-Service General Hospital, National Defense Medical Center, Taipei 11490, Taiwan; a2241@mail.ndmctsgh.edu.tw; 5School of Nursing, National Defense Medical Center, Taipei 11490, Taiwan

**Keywords:** mental adjustment, psychological adjustment, emotional adjustment, mental adaptation, psychological adaptation, emotional adaptation, comprehensive quality of life outcome, cancer patient, terminally ill patient, path modeling

## Abstract

Using path modeling, this study aimed to explore whether mental adjustment was directly or indirectly related to comprehensive quality of life outcome (CoQoLO) among patients with terminal cancer. We conducted a cross-sectional designed study among patients with terminal cancer who underwent convenience sampling at our northern Taiwan clinic from August 2019 to August 2020. Patient characteristics data were collected via structured questionnaires, namely, the Mini-Mental Adjustment to Cancer Scale and the Comprehensive Quality of Life Outcome Inventory. Descriptive statistics and regression analyses were used to examine the relationship between mental adjustment and CoQoLO. Path analysis described the dependencies among variables. For the 117 enrolled patients analyzed, MAC (β = 1.2, 95% confidence interval (CI) = 0.8–1.6, *p* < 0.001) and living with others (β = 19.9, 95% CI = 4.1–35.7, *p* = 0.015) were significant predictors and correlated positively with a CoQoLO score. Path modeling showed that the patients’ mental adjustment, economic status, perceived disease severity, palliative prognostic index, and symptom severity directly affected their CoQoLO. Our results indicate that the higher the mental adjustment, the better the CoQoLO among patients with terminal cancer. Thus, nurses need to assess mental adjustment levels when patients are hospitalized and accordingly develop interventions to improve the terminally ill patients’ mental adjustment to the final stages of cancer, thereby helping them to achieve good CoQoLO.

## 1. Introduction

Cancer remains the leading cause of death worldwide. In Taiwan, for example, the disease has been the foremost cause of death for more than 35 years, and approximately 50,000 cancer-related deaths occurred in 2018 [1]. In 2020, the number of cancer-related estimated deaths in the United States alone was 606,520 [2]. Despite medical advancements in cancer treatment and prolonged survival time, many patients must face the final stages of the disease and death. Terminally ill patients refer to those with serious injury or illness who are diagnosed by a physician as incurable and medical evidence regarding whom indicates a fatal prognosis in the near future [3]. Under the influence of traditional Chinese culture, most terminally ill patients cannot fully express their wishes or talk about death. This may cause patients to encounter multiple losses that may influence their psychological and spiritual well-being when dealing with the terminal stage [4]. A prior study indicated that patients with mental illness were associated with lower satisfaction of care [5]. It is necessary to support patients in their adaptation to the final stages of the disease to enable them to control their condition, find meaning in their lives, retain their self-esteem, and feel respected until death [6].

Mental adjustment is indicated by the coping responses involved in adopting strategies to deal with or adapt to a situation, such as severe and life-threatening medical conditions [7]. There is a set of cognitive and behavioral attempts to manage specific internal and external challenges that arise from situations of stress that are perceived as a burden or are beyond a person’s control [7]. Different types of mental adjustment strategies or coping responses develop as a result of patients’ personality traits, social contexts, and their families and health care professionals’ attitudes. Additionally, mental adjustment is affected by patient age, personality traits, religious attitudes, social networks, family support, and healthcare professionals [8]. Mental adjustment to cancer (MAC) could be maladaptive (e.g., helplessness–hopelessness (H&H) and anxious preoccupation (AP)) or adaptive (e.g., fighting spirit (FS)) [9]. H&H refers to the patients’ negative responses to their condition and disbelief in loss of control over circumstances [10]. AP indicates that patients could be anxious about their illness [11]. FS is defined as an optimistic attitude with a realistic appraisal of illness [10,11]. Studies indicate that maladaptive mental adjustment is an essential determinant of psychological morbidities, such as anxiety and depression [12] and poor psychological outcomes [13,14], which affect the quality of life (QoL) [8,15] and death [16].

The goal of palliative care is to assist terminally ill patients to achieve a good comprehensive quality of life outcome (CoQoLO), as well as a good death in the end stage. It includes the patient’s physical, psychological, social, and spiritual well-being. It encompasses minimal suffering and burden, awareness, preparedness and acceptance of death, a sense of closure, family and interpersonal wellness, emotional wellness, being seen and perceived as a person, exhibiting respect for the patient’s wishes, religious and spiritual wellness, and transparent decision-making [17,18]. Preparing for a good CoQoLO is necessary in Chinese culture as relatives and close contacts are expected to make efforts to address any unfinished business of persons on the verge of death. Many previous studies have suggested that mental adjustment is correlated with QoL [19,20] and spiritual well-being [21] in palliative care settings. A study also indicated that CoQoLO was associated with a reduced risk of anxiety or depression [16]. However, these studies only applied multiple regression models to identify the relationship between mental adjustment and QoL and did not conduct a path analysis to identify the indirect and direct effects of variables on this relationship. Although regression analyses play an essential role in providing evidence of factors that predict CoQoLO, path analysis helps one observe direct and indirect effects concurrently with multiple independent and dependent variables [22]. We hypothesized that mental adjustment may be correlated with CoQoLO among patients with terminal cancer, and there exist independent factors that directly or indirectly affect good death through mental adjustment. Therefore, this study aimed to explore the relationship between mental adjustment and CoQoLO using regression analysis and examine the direct or indirect relationship between mental adjustment and CoQoLO scores, using path modeling, among patients with terminal cancer.

## 2. Methods

### 2.1. Study Design and Population

We conducted a cross-sectional study that enrolled terminally ill cancer patients using convenience sampling from the hematology and oncology wards or acute wards of medical centers in northern Taiwan from August 2019 to August 2020. A sample size of 109 was required to perform the F test of multiple linear regression analysis (using G* power 3.1.9.2) [23]. We assigned a *p* value of 0.05, power of 0.80, and effect size of 0.19. Considering an 80% response rate of the survey, we originally recruited 137 participants diagnosed with terminal cancer, aged > 18 years, free of cognitive impairments or mental illness, who were able to communicate in Mandarin or Taiwanese, and who agreed to participate in this study. Of these, 20 participants either rejected the survey or were too weak to complete it, and thus, were excluded from the study. Finally, 117 participants completed the questionnaires, as described below.

### 2.2. Measurements

#### 2.2.1. Characteristics of Patients with Terminal Cancer

Patient demographics and clinical data included age, sex, educational level, religious belief, marital status, living with others (yes or no), having caregivers (yes or no), economic source (self or others), economic status (<NTD 20,000 or ≥NTD 20,000), comorbidities (yes or no), perceived disease severity (Likert scale 1–5, 1 = not serious at all, 5 = very serious), cancer diagnosis, palliative prognostic index (PPI), and symptom severity.

PPI was developed by Morital et al. [24] and is used to predict the survival time of terminally ill cancer patients. It contains five items: performance status, oral intake, edema, dyspnea at rest, and delirium. The total PPI score ranges from 0 to 15, with a lower score indicating a longer survival time. The index is shown to predict the survival time of terminally ill patients with high sensitivity and specificity [24].

Symptom severity was measured using the Memorial Symptom Assessment Scale Short Form (MSAS-SF). This scale was originally developed by Portenoy et al. [25], refined with a short-form version by Chan et al. [26], and translated into Mandarin for the Chinese population [27]. The scale records the patients’ self-reported severity of 28 physical and four psychological symptoms during the past week. Each symptom severity was rated on a five-point scale, with 0 indicating “no symptoms” and 4 indicating “very severe symptoms”. The total scores range from 0 to 128, with a higher score representing more severe symptoms. Good validity and reliability have been reported in patients with cancer [27]. In this study, Cronbach’s α was 0.892 for 117 patients with terminal cancer.

#### 2.2.2. Mini-Mental Adjustment to Cancer Scale (Mini-MAC)

Mini-MAC was developed by Waston et al. [9] and translated into different languages [10,28,29,30,31], including Mandarin [32]. It is an instrument to record self-reported coping response of patients with cancer. It contains 29 items and is divided into five dimensions: H&H (8-item), AP (8-item), FS (4-item), cognitive avoidance (CA, 4-item), and fatalism (FA, 5-item). Each item is rated on a four-point numeric scale, ranging from 0 to 3, where 0 represents “definitely does not apply to me”, and 3 represents “definitely applies to me” H&H, AP, and CA correspond to the more passive coping strategies, whereas FS and FA represent the more active coping strategies. We then transformed the passive coping strategies (AP, H&H, and CA) into positive items to calculate the total scores for mental adjustment. The total scores ranged from 0 to 87, with higher scores indicating higher mental adjustment to the terminal stage. Mini-MAC has demonstrated good reliability and validity in patients with cancer [9], including terminally ill cancer patients [10]. In this study, the internal consistency of Cronbach’s α for the Mini-MAC was 0.853, and ranged from 0.730 to 0.872 in these five dimensions for 117 patients with terminal cancer.

#### 2.2.3. Comprehensive Quality of Life Outcome Inventory

The Comprehensive Quality of Life Outcome (CoQoLO) inventory is used to assess whether a patient with advanced cancer will achieve a good death [33]. The scale is self-reported and contains 28 items, which are divided into 10 subscales (i.e., physical and psychological comfort, staying in a favorite place, maintaining hope and pleasure, good relationships with medical staff, not being a burden to others, good relationships with family, independence, environmental comfort, being respected as an individual, and having a fulfilling life). Each item is rated on a seven-point Likert scale, with 1 representing “completely disagree” and 7 representing “completely agree”. The total scores of good death levels range from 28 to 196, with a higher score indicating a good death level. This inventory has been revealed to have good validity and reliability in a sample of 405 cancer patients [34]. In the present study, Cronbach’s α was 0.928 for 117 patients with terminal cancer.

### 2.3. Study Process

The study was conducted after receiving approval from the pertinent institutional review board (approval no. 2–108–05–029). Patients with terminal cancer who fulfilled the inclusion criteria were provided with information on the study protocol. The researcher described the objectives and methods of the study to participants in a conversation room. After written informed consent was obtained from each participant, the researcher collected data via questionnaires through an interview. The questionnaires were anonymous, and the collected information was considered confidential. Participants spent 15–20 min completing the questionnaires and were entitled to discontinue the study at any time. Following completion of the questionnaires, each participant received a gift card.

### 2.4. Statistical Analysis

Data were analyzed using IBM SPSS Statistics for Windows, Version 22.0 (IBM Corp., Released 2013. Armonk, NY, USA). Categorical variables were described as frequencies and percentages. Continuous variables were expressed as mean ± standard deviation (±SD). Pearson’s correlation was used to analyze the relationship between mental adjustment and CoQoLO along with its dimensions. Multiple linear regression was used to determine the predictors of good death. Furthermore, we used path analysis to describe the direct or indirect dependencies among a set of variables, including patient characteristics. Statistical significance was set at *p* < 0.05.

## 3. Results

The mean age of the participants was 57.4 years. Most of the patients were women (59.8%), with >9 years of education (70.9%). The majority of them had religious beliefs (84.6%), were married (69.2%), lived with others (90.6%), had caregivers (91.5%), had good economic sources from others (77.8%), earned > NTD 20,000 per month, had no comorbidities (55.6%), and had non-breast cancers (lung cancers, colorectal cancers, head and neck cancer, etc.) (70.1%). The mean scores of perceived severity, PPI, and symptom severity were 3.3, 1.5, and 22.0 points, respectively (Table 1).

The mean CoQoLO score was 145.2 points (±SD = 25.8). The mean score for mental adjustment was 49.9 points (±SD = 11.0). The mean scores for dimensions of mental adjustment, H&H, AP, FS, CA, and FA were 8.6 points (±SD = 5.1), 12.5 points (±SD = 5.1), 8.7 points (±SD = 2.0), 6.8 points (±SD = 2.4), and 9.1 points (±SD = 2.5), respectively. Mental adjustment was significantly and positively correlated with the CoQoLO score (r = 0.589, *p* < 0.001). As mental adjustment increased, the CoQoLO score improved among patients with terminal cancer. Next, we transformed the negative subscales (H&H, AP, and CA) into positive items. The results showed that H&H (r = −0.570, *p* < 0.001), AP (r = −0.549, *p* < 0.001), and FS (r = 0.363, *p* < 0.001) dimensions were significantly correlated with CoQoLO. Higher levels of H&H and AP had a poorer CoQoLO scores, and a higher FS indicated a better CoQoLO score among patients with terminal cancer (Table 2).

As shown in Table 3, mental adjustment and living with others were significant predictors of a good CoQoLO score among the patients after adjustment for patient characteristics. A patient with a higher mean score for mental adjustment had a better CoQoLO score (β = 1.2, 95% CI = 0.8–1.6, *p* < 0.001). Patients who lived with others had a higher mean CoQoLO score than those who did not live with others, and thus, had a lower mean score (β = 19.9 points; 95% CI = 4.1–35.7, *p* = 0.015).

Path modeling demonstrated that mental adjustment, economic status, perceived disease severity, PPI, and symptom severity directly affected CoQoLO. The relationships between economic status more than NTD 20,000 per month compared with less than NTD 20,000 per month (coefficients = 0.233, *p* = 0.014), perceived disease severity (coefficients = −0.306, *p* < 0.001), PPI (coefficients = −0.344, *p* < 0.001), symptom severity (coefficients = −0.401, *p* < 0.001) and mental adjustment (coefficients = 0.594, *p* < 0.001) were significant by standardized coefficient estimates for the paths. The results are shown in Table 4.

## 4. Discussion

To the best of our knowledge, this study is one of the few to explore the relationship between mental adjustment and CoQoLO among patients with terminal cancer by using patient-reported outcomes. In addition to regression analysis, this study also used path analysis to examine the direct or indirect relationship between mental adjustment and CoQoLO scores among patients with terminal cancer. We found that patients with terminal cancer who were better mentally adjusted had better CoQoLO. This finding agrees with those of previous studies [15,19,34,35]. Indeed, successful mental adjustment to a life-threatening event depends on how an individual with an experience of living with cancer or even terminally ill cancer attempts to retrieve or gain control over the event and over life, and attempts to reestablish self-esteem, along with a positive attitude toward life [36]. A better mental adjustment indicates less depression, less anxiety, and greater life satisfaction [13]. It could expand people’s thought–action repertoires and establish resources for coping, which could significantly facilitate good CoQoLO.

This study showed that terminally ill cancer patients with a higher level of H&H, and AP resulted in poorer CoQoLO, whereas patients with a higher FS had a better CoQoLO, consistent with prior studies [15,34]. Patients encountering H&H displayed discreteness, withdrawal, and reduced effort to deal with the situation that generated stress or distress, and hence, these emotions (H&H) were unfavorable to the adaptive process [10,11]. This adverse condition correlated negatively with patients’ physical, emotional, and functional well-being, health outcomes, and CoQoLO [15,19]. AP, as a cancer-specific coping response, could be harmful to the well-being of patients with terminal cancer, thereby affecting their emotions and attitudes toward dealing with the end stage [11]. This could negatively impact how they enjoy life and lessen their appreciation of the positive characteristics of existence, and, in turn, make them feel more anxious about death, leading to maladjustment. Their AP may direct them to foster other negative emotions and thought patterns, resulting in poor death. FS is a set of cognitive and behavioral efforts to fight back and face the illness head-on to enhance well-being [10,11]. It has also been shown to form the basis of mental adjustment, essential to achieving a fulfilling life, and supporting patients to hold on until their death [37]. Patients with a high FS view the disease as a challenge, have a positive vision of the future, and believe in making the necessary efforts to control the illness [38]. Therefore, in this context, the FS of terminally ill cancer patients could be a protective attribute against negative emotional experiences and thus improve their CoQoLO.

This study found that patients with terminal cancer who lived with others had significantly better CoQoLO than those who did not live with others. Studies have shown that patients with terminal cancer who perceive higher support from others are more likely to report better death preparedness [39]. Family members strive for balance and well-being to accompany their affected relatives by helping them find meaning and strength in their end-of-life stages and regain balance and harmony toward the end of their lives [40]. In addition, patients living with others can obtain the latter’s support in saying goodbye, opine on euthanasia in case of unbearable suffering, and receive help to achieve a good CoQoLO [41]. In this path analysis, we also found that living with others indirectly affected good CoQoLO through mental adjustment. Patients’ perceived social support provides a protective factor for mental adjustment [8]. An earlier study showed that patients with better social support demonstrated better mental adjustment [13], aided by mental adjustments of close relatives or families in dealing with patients with terminal cancer and the possible death of their loved ones. Some of these concepts include believing death as a distant possibility, trusting that the disease would improve, encouraging people to live in the moment, and making good use of family support and social networks [42]. Taiwan is a family-centered country, and terminally ill patients need their family members to provide care, including generous financial, and emotional support, thus improving the patients’ CoQoLO.

This study indicated that patients’ mental adjustment, economic status, perceived disease severity, PPI, and symptom severity directly affect their CoQoLO. It is not surprising that patients’ or families’ economic status would facilitate good CoQoLO [43] due to higher spiritual burden [44]. The perception of one’s disease severity, and not the actual disease state, tended to relate to a patient’s psychological well-being. A prior study has confirmed that patients’ subjective perception of disease severity was correlated with psychological well-being or QoL [45]. However, these studies exclusively included women with urinary incontinence and congenital heart disease and did not focus on patients with terminal cancer. A patient’s psychological state, for example, their self-perceived burden, is thought to be a universal concern across countries and is important for achieving a good CoQoLO [46]. Terminally ill patients with cancer experience a self-perceived burden that often affects their well-being and causes profound suffering that is associated with hopelessness, reduced CoQoLO, and depression, as well as becomes a barrier to the optimal achievement of good CoQoLO [47]. Therefore, it is crucial to monitor patients’ perception of disease severity, PPI, or symptom severity to improve their psychological well-being at the terminal stages of their lives to achieve good CoQoLO.

This study has several limitations. First, it was a cross-sectional study; thus, a longitudinal follow-up study is needed to confirm a causal relationship between mental adjustment and CoQoLO. Second, the participants were all patients with terminal cancer and were hospitalized; hence, since the study did not include outpatients, the results may present an overestimation of patients’ disease severity. Third, the convenience sampling approach was utilized to enroll hospitalized terminally ill cancer patients in acute wards, not in terminally ill non-cancer patients or critical inpatients. The generalizability of the findings may be limited to terminally ill cancer inpatients in the acute wards of a medical center. Our findings have several implications. First, they provide the foundation for future studies to explore longitudinal relationships between mental adjustment and CoQoLO among patients with terminal cancer. Second, future study populations may include outpatients with terminal cancer to compare the differences between mental adjustment and CoQoLO among terminal inpatients and outpatients. Third, random sampling should be used to avoid selection bias and improve the generalizability of the findings.

## 5. Conclusions

Our results indicated that the higher the mental adjustment, the better the CoQoLO among patients with terminal cancer. Patients who lived with others were more likely to have a better CoQoLO than those who did not live with others. Patients with better mental adjustment, high economic status, low perception of disease severity, low PPI, and low symptom severity had an improved CoQoLO. Similarly, living with others contributed to better mental adjustment and CoQoLO. Practically, we recommend assessing patients’ mental adjustment levels when they are hospitalized and designing future interventions that focus on enhancing mental adjustment among terminal cancer patients to help them achieve a good CoQoLO.

## Figures and Tables

**Table 1 ijerph-18-04926-t001:** Characteristics of terminally ill cancer patients (N = 117).

Variables	Mean + SD/N (%)
Age	57.4 ± 11.3
Gender	
Male	47 (40.2%)
Female	70 (59.8%)
Educational level	
<9 years	34 (29.1%)
>=9 years	83 (70.9%)
Religious belief	
No	18 (15.4%)
Yes	99 (84.6%)
Marital status	
Unmarried	36 (30.8%)
Married	81 (69.2%)
Living with others	
No	11 (9.4%)
Yes	106 (90.6%)
Having caregivers	
No	10 (8.5%)
Yes	107 (91.5%)
Economic sources	
Self	26 (22.2%)
Others	91 (77.8%)
Economic status	
< NTD 20,000	56 (47.9%)
≥ NTD 20,000	61 (52.1%)
Comorbidities	
No	65 (55.6%)
Yes	52 (44.4%)
Perceived disease severity	3.3 ± 1.0
Cancer diagnosis	
Breast cancer	35 (29.9%)
Non-breast cancer	82 (70.1%)
Palliative prognostic index	1.5 ± 2.4
Symptom severity	22.0 ± 11.3

**Table 2 ijerph-18-04926-t002:** Correlation of mental adjustment and its dimensions with comprehensive quality of life outcome (CoQoLO) among terminally ill cancer patients (N = 117).

Variables	Mean + SD	r	*p*-Value
CoQoLO	145.2 ± 25.8		
Mental adjustment	49.9 ± 11.0	0.589	<0.001
Helpless and Hopeless	8.6 ± 5.1	−0.570	<0.001
Anxious Preoccupation	12.5 ± 5.1	−0.549	<0.001
Fighting Spirit	8.7 ± 2.0	0.363	<0.001
Cognitive Avoidance	6.8 ± 2.4	−0.034	0.719
Fatalism	9.1 ± 2.5	0.010	0.916

**Table 3 ijerph-18-04926-t003:** Predictors of comprehensive quality of life outcome (CoQoLO) among terminally ill cancer patients (N = 117).

Independent Variables	Crude β (95% CI)	*p*-Value	Adjusted β (95% CI)	*p*-Value
Mental adjustment	1.4 (1.0–1.7)	<0.001	1.2 (0.8–1.6)	<0.001
Age	0.01 (−0.4–0.4)	0.955	0.2 (−0.1–0.6)	0.229
Gender				
Male	Reference		Reference	
Female	1.1 (−8.5–10.7)	0.817	0.8 (−8.4–9.9)	0.872
Educational level				
<9 years	Reference		Reference	
>=9 years	−1.9 (−12.3–8.4)	0.718	−4.4 (−13.2–4.4)	0.333
Religious belief				
No	Reference		Reference	
Yes	−4.1 (−17.1–9.0)	0.542	1.2 (−9.4–11.7)	0.831
Marital status				
Unmarried	Reference		Reference	
Married	7.8 (−2.3–17.8)	0.135	2.4 (−6.6–11.5)	0.596
Living with others				
No	Reference		Reference	
Yes	14.3 (−1.6–30.2)	0.081	19.9 (4.1–35.7)	0.015
Having caregivers			0.653	
No	Reference		Reference	
Yes	3.9 (−12.9–20.7)	0.653	−6.7 (−23.9–10.5)	0.445
Economic sources			0.643	
Self	Reference		Reference	
Others	−2.7 (−14.0–8.6)	0.644	1.9 (−7.7–11.6)	0.698
Economic status				
< NTD 20,000	Reference		Reference	
> NTD 20,000	12.0 (2.8–21.1)	0.012	6.2 (−1.6–14.0)	0.123
Comorbidities				
No	Reference		Reference	
Yes	−7.7 (−17.0–1.7)	0.110	−3.2 (−11.2–4.9)	0.441
Perceived disease severity	−8.8 (−13.1–4.4)	<0.001	−2.6 (−6.6–1.3)	0.193
Cancer diagnosis				
Breast cancer	Reference		Reference	
Non-breast cancer	−2.7 (−12.9–7.6)	0.609	1.9 (−7.9–11.7)	0.702
Palliative prognostic index	−3.9 (−5.7–−2.0)	<0.001	−1.7 (−4.0–0.5)	0.140
Symptom severity	−0.9 (−1.3–−0.5)	<0.001	−0.1 (−0.6–0.3)	0.557

**Table 4 ijerph-18-04926-t004:** Path modeling for mental adjustment and comprehensive quality of life outcome (CoQoLO) among terminally ill cancer patients (N = 117).

Variables	Mental Adjustment→Good Death		Good Death	
	Coefficients	*p*-Value	Coefficients	*p*-Value
Mental adjustment			0.594	<0.001
Age	0.052	0.578	0.005	0.957
Gender				
Female/Male	−0.008	0.934	0.022	0.814
Educational level				
≥9 years/<9 years	−0.053	0.571	−0.034	0.716
Religious belief				
Yes/No	0.007	0.944	−0.057	0.542
Marital status				
Married/Unmarried	0.093	0.322	0.139	0.139
Living with others				
Yes/No	0.151	0.108	0.162	0.085
Having caregivers				
Yes/No	0.050	0.591	0.042	0.653
Economic sources				
Others/Self	0.045	0.630	−0.043	0.646
Economic status				
≥NTD 20,000 <NTD 20,000	0.076	0.418	0.233	0.014
Comorbidities				
Yes/No	−0.029	0.754	−0.148	0.115
Perceived disease severity	−0.074	0.432	−0.344	<0.001
Cancer diagnosis				
Non-breast cancer/Breast cancer	0.025	0.792	−0.048	0.608
Palliative prognostic index	−0.114	0.224	−0.361	<0.001
Symptom severity	−0.119	0.205	−0.401	<0.001

## Data Availability

The data presented in this study are available on request from the corresponding author. The data are not publicly available due to privacy.

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
