# Peer review of "Mental Adjustment as a Predictor of Comprehensive Quality of Life Outcome among Patients with Terminal Cancer"

_ijerph, 2021, doi:10.3390/ijerph18094926_

Round 1

Reviewer 1 Report

I really appreciated this study! It is very important and interesting! I send the file with some little suggestions.

Author Response

Reviewer1#

Comments and Suggestions for Authors

I really appreciated this study! It is very important and interesting! I send the file with some little suggestions.

Authors’ response:

Thank you very much.

Abstract

“…examine the causal relationship.” Only experimental studies allow establishing causal relationships. As this is an observational study, I suggest reviewing this sentence.

Authors’ response:

Thank you very much for the suggestions. We have reviewed the sentence and revised to “Using path modeling, this study aimed to explore whether mental adjustment was directly or indirectly related to comprehensive quality of life outcome (CoQoLO) among patients with terminal cancer.” in Abstract section.

Methods

I suggest to report this article in according the STROBE checklist.

Authors’ response:

Thank you very much for the suggestion. We have reported this article according to the STROBE checklist.

Was the data collection carried out through an interview?

Authors’ response:

Thank you. “…, the researcher collected data via questionnaires through an interview.” We have added in the revised manuscript and using “Track Changes”.

Reviewer 2 Report

I would like to start by congratulating the authors and congratulating them for having decided to investigate an area where there is still so much to be discovered, but also for having decided to share this protocol with the rest of the scientific community, so that science can evolve.

This is a cross-sectional designed study on Mental Adjustment to Cancer as a Predictor of Good Death Among Patients with Terminal Cancer.

All comments, doubts and suggestions presented are in the constructive sense and to try to improve the article, after several attentive readings.

Abstract

I would like to see a conclusion more focused on clinical application.

Keywords:

Repetitions with expressions that are in the title should be avoided. Whenever possible, keywords should be Mesh.

Introduction

The objective defined at the end of the Introduction must be exactly the same as the Abstract or vice versa.

The relevance and innovation of this study is not clear.

Materials and methods

It is not clear how authors determined the sample size.

It is unclear how authors selected the sample.

It is not clear the potential benefit for these patients of completing these extensive questionnaires at a terminal stage of their lives.

Discussion and Conclusions

Authors present the limitations of the study in a very honest and pragmatic way.

I would have liked to read a conclusion more directed to clinical practice and how this study may or may not help to change the practices implemented.

General comments

The data presented are from August 2020, more than 7 months ago.

Exactly because of the limitations authors present, I am of the opinion that this article should not be accepted in this journal. But I believe that the authors should be encouraged to resubmit the article with some corrections already made to others in MDPI.

The focus should always be the real interest for clinical practice. What was the purpose of this study for the studied population or for similar populations? The last sentence of the conclusion is insufficient to answer this question.

Author Response

Reviewer2#

Comments and Suggestions for Authors

I would like to start by congratulating the authors and congratulating them for having decided to investigate an area where there is still so much to be discovered, but also for having decided to share this protocol with the rest of the scientific community, so that science can evolve.

This is a cross-sectional designed study on Mental Adjustment to Cancer as a Predictor of Good Death Among Patients with Terminal Cancer. All comments, doubts and suggestions presented are in the constructive sense and to try to improve the article, after several attentive readings.

Abstract

I would like to see a conclusion more focused on clinical application.

Authors’ response:

Thank you very much for the comment. We have revised in the conclusion of Abstract and Conclusion section using “Track Changes”.

Keywords:

Repetitions with expressions that are in the title should be avoided. Whenever possible, keywords should be Mesh.

Authors’ response:

Thank you very much. We have deleted the repetition word in the title and added the Mesh term.

Introduction

The objective defined at the end of the Introduction must be exactly the same as the Abstract or vice versa. The relevance and innovation of this study is not clear.

Authors’ response:

Thank you very much for the comments. We have revised in Abstract and Introduction to emphasize the relevance and innovation of this study.

Materials and methods

It is not clear how authors determined the sample size. It is unclear how authors selected the sample. It is not clear the potential benefit for these patients of completing these extensive questionnaires at a terminal stage of their lives.

Authors’ response:

Thank you very much for the comments. We have described the sample size and how authors selected the sample in “Study design and population”. We also described the potential benefit for these patients of completing these extensive questionnaires in Study process.

Discussion and Conclusions

Authors present the limitations of the study in a very honest and pragmatic way. I would have liked to read a conclusion more directed to clinical practice and how this study may or may not help to change the practices implemented.

Authors’ response:

Thank you very much for the suggestions. We have revised in the conclusion of Abstract and Conclusion section using “Track Changes”.

General comments

The data presented are from August 2020, more than 7 months ago. Exactly because of the limitations authors present, I am of the opinion that this article should not be accepted in this journal. But I believe that the authors should be encouraged to resubmit the article with some corrections already made to others in MDPI. The focus should always be the real interest for clinical practice. What was the purpose of this study for the studied population or for similar populations? The last sentence of the conclusion is insufficient to answer this question.

Authors’ response:

Thank you very much for the comments. “this study aimed to explore the relationship between mental adjustment and CoQoLO using regression analysis, and examine the direct or indirect relationship between mental adjustment and CoQoLO scores, using path modeling, among patients with terminal cancer.” We have pointed out in last paragraph of Introduction. We also revised the Conclusion section to answer the objective of this study. Our results indicated that the higher the mental adjustment, the better the CoQoLO among patients with terminal cancer. Patients who lived with others were more likely to have a better CoQoLO than those who did not live with others. Patients with better mental adjustment, high economic status, low perception of disease severity, low PPI, and low symptom severity had an improved CoQoLO. Similarly, living with others contributed to better mental adjustment and CoQoLO. Practically, we recommend assessing patients’ mental adjustment levels when they are hospitalized and designing future interventions that focus on enhancing mental adjustment among terminal cancer patients to help them achieve a good CoQoLO.

Reviewer 3 Report

Major comments:

The reviewed paper proposes an interesting problem, which is not only theoretically, but could be also practically important.  Author(s) focused on explore the relationship between MAC and good death and examine the causal relationship among patients with terminal cancer using path modeling. Author(s) taken into account descriptive statistics and regression analysis to examine the relationship between Mental Adjustment to Cancer (MAC) and good death. Author(s) considered methods that can help the developing interventions to improve terminal patients' mental adjustment could help them achieve good death.   The paper has a logical structure and is clearly, concisely and accurately written.  I suggest to update abstract to highlight most important findings of this research. The “Introduction“ part should be updated, the author(s) did not clearly show the difference between their approach and those in the literature. Complex and expanded state-of-the-art is needed. I also suggest author(s) should add discussion about pros and cons of considered problem to clearly identify the benefits of the introduced approach. The findings are too much dependent on the used approach and the case study to be generalized.  I suggest to add all appropriate references from the list below (https://doi.org/):

10.1186/s12888-021-03071-y

10.1097/MD.0000000000003716
10.1186/s12913-020-05294-3

10.1513/AnnalsATS.201806-391OC

Minor comments:  

Paper contains some amount of typos that need to be corrected throughout the paper. There are several minor language errors in the text. Some sentences require rewriting. Some acronyms were not defined. 

Author Response

Reviewer3#

Major comments:

The reviewed paper proposes an interesting problem, which is not only theoretically, but could be also practically important. Author(s) focused on explore the relationship between MAC and good death and examine the causal relationship among patients with terminal cancer using path modeling. Author(s) taken into account descriptive statistics and regression analysis to examine the relationship between Mental Adjustment to Cancer (MAC) and good death. Author(s) considered methods that can help the developing interventions to improve terminal patients' mental adjustment could help them achieve good death. The paper has a logical structure and is clearly, concisely and accurately written. 

Authors’ response:

Thank you very much.

I suggest to update abstract to highlight most important findings of this research. The “Introduction“ part should be updated, the author(s) did not clearly show the difference between their approach and those in the literature. Complex and expanded state-of-the-art is needed. I also suggest author(s) should add discussion about pros and cons of considered problem to clearly identify the benefits of the introduced approach. The findings are too much dependent on the used approach and the case study to be generalized.  I suggest to add all appropriate references from the list below (https://doi.org/):

10.1186/s12888-021-03071-y

10.1097/MD.0000000000003716
10.1186/s12913-020-05294-3

10.1513/AnnalsATS.201806-391OC

Authors’ response:

Thank you very much for the suggestions. We have added the appropriate references from the list above in the Introduction section.

Minor comments:  

Paper contains some amount of typos that need to be corrected throughout the paper. There are several minor language errors in the text. Some sentences require rewriting. Some acronyms were not defined. 

Authors’ response:

Thank you very much. This manuscript also has been revised by English expert.

Reviewer 4 Report

Dear authors,

I am glad to review your manuscript and interesting work here. Thank you for letting me read and review. 

I think your work is publishable after minor revisions:

The abstract is complete and concise. 

The introduction is fine, but maybe could content a descriptive analysis of the studied population, in order to offer the readers a cultural and contextual framework to better understand the results and the complete report.  

At Methods section:

About G*power:

https://www.psychologie.hhu.de/arbeitsgruppen/allgemeine-psychologie-und-arbeitspsychologie/gpower 

If you use G*Power for your research, then we would appreciate your including one or both of the following references (depending on what is appropriate) to the program in the papers in which you publish your results:

Faul, F., Erdfelder, E., Buchner, A., & Lang, A.-G. (2009). Statistical power analyses using G*Power 3.1: Tests for correlation and regression analyses. Behavior Research Methods41, 1149-1160. Download PDF

Cite SPSS with year, like IBM Corp. Released 2013. IBM SPSS Statistics for Windows, Version 22.0. Armonk, NY: IBM Corp

I would like to ask about religious beliefs and the better good death, it could be desirable a deeper discussion in this key topic. Reference number 13 is so interesting and would help enhance the wealth of information, insights, and the spectrum of perspectives that shows readers a new perspective about preparation for a good death:   https://pubmed.ncbi.nlm.nih.gov/31648531/ 

Bovero A, Gottardo F, Botto R, Tosi C, Selvatico M, Torta R. Definition of a Good Death, Attitudes Toward Death, and Feelings of Interconnectedness Among People Taking Care of Terminally ill Patients With Cancer: An Exploratory Study. Am J Hosp Palliat Care. 2020 May;37(5):343-349. DOI: 10.1177/1049909119883835. Epub 2019 Oct 24. PMID: 31648531.

Author Response

Reviewer4#

Dear authors,

I am glad to review your manuscript and interesting work here. Thank you for letting me read and review. 

Authors’ response:

Thank you very much.

I think your work is publishable after minor revisions:

The abstract is complete and concise. 

Authors’ response:

Thank you very much.

The introduction is fine, but maybe could content a descriptive analysis of the studied population, in order to offer the readers a cultural and contextual framework to better understand the results and the complete report.  

Authors’ response:

Thank you very much for your valuable comments. We have described the studied population and used Track Change in the Introduction section.

At Methods section:

About G*power:

https://www.psychologie.hhu.de/arbeitsgruppen/allgemeine-psychologie-und-arbeitspsychologie/gpower 

If you use G*Power for your research, then we would appreciate your including one or both of the following references (depending on what is appropriate) to the program in the papers in which you publish your results: Faul, F., Erdfelder, E., Buchner, A., & Lang, A.-G. (2009). Statistical power analyses using G*Power 3.1: Tests for correlation and regression analyses. Behavior Research Methods41, 1149-1160. Download PDF

Authors’ response:

Thank you very much. We have cited the paper in 2.1 Study design and population.

Cite SPSS with year, like IBM Corp. Released 2013. IBM SPSS Statistics for Windows, Version 22.0. Armonk, NY: IBM Corp

Authors’ response:

Thank you very much. We have revised the citation in 2.4 Statistical analysis.

I would like to ask about religious beliefs and the better good death, it could be desirable a deeper discussion in this key topic. Reference number 13 is so interesting and would help enhance the wealth of information, insights, and the spectrum of perspectives that shows readers a new perspective about preparation for a good death:   https://pubmed.ncbi.nlm.nih.gov/31648531/ 

Bovero A, Gottardo F, Botto R, Tosi C, Selvatico M, Torta R. Definition of a Good Death, Attitudes Toward Death, and Feelings of Interconnectedness Among People Taking Care of Terminally ill Patients With Cancer: An Exploratory Study. Am J Hosp Palliat Care. 2020 May;37(5):343-349. DOI: 10.1177/1049909119883835. Epub 2019 Oct 24. PMID: 31648531.

Authors’ response:

Thank you very much for your valuable comments. “The goal of palliative care is to assist terminally ill patients to achieve a good comprehensive quality of life outcome (CoQoLO) as well as good death in the end stage. It includes the patient’s physical, psychological, social, and spiritual well-being. It encompasses minimal suffering and burden, awareness, preparedness and acceptance of death, a sense of closure, family and interpersonal wellness, emotional wellness, being seen and perceived as a person, exhibiting respect for the patient’s wishes, religious and spiritual wellness, and transparent decision-making [17, 18]” (The above had mentioned in the last paragraph of Introduction). We explored the factors related to CoQoLO, such as religious belief, not only using regression analysis, but also using path model to examine the direct and indirect effects concurrently with multiple independent and dependent variables. The results showed that religious belief was not significant predictor of CoQoLO in regression analysis (β = 1.2, 95% CI = -9.4 - 11.7, p < 0.831). In addition, religious belief did not indirectly (coefficients = 0.007, p = 0.944) or directly (coefficients = -0.057, p = 0.542) affect CoQoLO. Therefore, we did not have a deeper discussion in this topic.

  1. Krikorian, A.; Maldonado, C.; Pastrana, T., Patient's Perspectives on the Notion of a Good Death: A Systematic Review of the Literature. J Pain Symptom Manage. 2020, 59, (1), 152-164.
  2. Bovero, A.; Gottardo, F.; Botto, R.; Tosi, C.; Selvatico, M.; Torta, R., Definition of a good death, attitudes toward death, and feelings of interconnectedness among people taking care of terminally ill patients with cancer: an exploratory study. Am J Hosp Palliat Care. 2020, 37, (5), 343-349.

Reviewer 5 Report

In this study, the authors explored the relationship between mental adjustment to cancer and good death.

Suggestions and questions (answers can/should be used to improve the paper):
1. The paragraph started by "As we know, the ultimate goal..." (Introduction) should be split.
2. What are the research questions of the study?
3. The term 'fighting spirit' could be better explained/exemplified at the first occurrences, not in the discussion section.
4. The sentence "the convenience sampling approach utilized may compromise the generalizability of the findings" is stated, but it is not enough to discuss this point. How generalized is the results? Different cultural and social aspects are considered in this study (e.g., religious beliefs, married). Do they influence results?
5. The relevance of the study is poorly described in the manuscript. Why (or for whom) are the results important?
6. What is the contribution of the study? It should be explicitly declared.
7. Conclusion is poor. The text should show that the objective was (or was not) achieved.

Specific comments:
- All abbreviations should be defined at the first occurrence (e.g., MAC, QoL, etc) including in the abstract.
- death.. -> death.
- "...and neck cancer, and etc.)" -> remove the second 'and'

Author Response

Reviewer5#

In this study, the authors explored the relationship between mental adjustment to cancer and good death.

Suggestions and questions (answers can/should be used to improve the paper):
1. The paragraph started by "As we know, the ultimate goal..." (Introduction) should be split.

Authors’ response:

Thank you very much. We have revised in the last paragraph of Introduction.

  1. What are the research questions of the study?

Authors’ response:

Thank you very much for your valuable comments. We had added the research question of the study in the last paragraph of Introduction.

  1. The term 'fighting spirit' could be better explained/exemplified at the first occurrences, not in the discussion section.

Authors’ response:

Thank you very much for the suggestions. We have explained the term ‘fighting spirit’ at the first occurrences in the second paragraph of Introduction.

  1. The sentence "the convenience sampling approach utilized may compromise the generalizability of the findings" is stated, but it is not enough to discuss this point. How generalized is the results? Different cultural and social aspects are considered in this study (e.g., religious beliefs, married). Do they influence results?

Authors’ response:

Thank you very much for the valuable comments. “The generalizability of the findings may be limited to terminally ill cancer inpatients in the acute wards of a medical center.” We have revised in the last paragraph of Discussion. In this study, we explored the factors related to CoQoLO, such as religious beliefs, married…and so on, not only using regression analysis, but also using path model to examine the direct and indirect effects concurrently with multiple independent and dependent variables. The results showed that mental adjustment and living with others were significant predictors of CoQoLO among the patients after adjustment for patient characteristics (Table 3). Path modeling demonstrated that mental adjustment, economic status, perceived disease severity, palliative prognostic index, and symptom severity directly affect CoQoLO (Table 4). We did not find the factors such as religious beliefs, or married that influence CoQoLO among terminally ill cancer patients.

  1. The relevance of the study is poorly described in the manuscript. Why (or for whom) are the results important?

Authors’ response:

Thank you very much for the comments. We have rechecked and revised the Introduction. In addition, we also pointed out the importance of this study.

  1. What is the contribution of the study? It should be explicitly declared.

Authors’ response:

Thank you very much. “Our results indicated that the higher the mental adjustment, the better the CoQoLO among patients with terminal cancer. Patients who lived with others were more likely to have a better CoQoLO than those who did not live with others. Practically, we recommend assessing patients’ mental adjustment levels when they are hospitalized and designing future interventions that focus on enhancing mental adjustment among terminal cancer patients to help them achieve a good CoQoLO” The above had revised and shown in Conclusion.

  1. Conclusion is poor. The text should show that the objective was (or was not) achieved.

Authors’ response:

Thank you very much for the comments. We have rechecked and revised the Conclusion to echo the objective of this study.

Specific comments:
- All abbreviations should be defined at the first occurrence (e.g., MAC, QoL, etc) including in the abstract.

Authors’ response:

We have defined all abbreviations. Thank you very much.

- death.. -> death.

Authors’ response:

We have revised. Thank you.

- "...and neck cancer, and etc.)" -> remove the second 'and'

Authors’ response:

We have removed the second “and”. Thank you.

Round 2

Reviewer 2 Report

The authors made important alterations to the article. It may be in conditions of being published.